# Immunosuppressive Signaling Pathways as Targeted Cancer Therapies

**DOI:** 10.3390/biomedicines10030682

**Published:** 2022-03-16

**Authors:** Botle Precious Setlai, Rodney Hull, Meshack Bida, Chrisna Durandt, Thanyani Victor Mulaudzi, Aristotelis Chatziioannou, Zodwa Dlamini

**Affiliations:** 1Department of Surgery, Faculty of Health Sciences, University of Pretoria, Private Bag X323, Arcadia 0007, South Africa; thanyani.mulaudzi@up.ac.za; 2SAMRC Precision Oncology Research Unit (PORU), DSI/NRF SARChI Chair in Precision Oncology and Cancer Prevention (POCP), Pan African Cancer Research Institute (PACRI), University of Pretoria, Hatfield 0028, South Africa; rodneyhull@gmail.com; 3Department of Anatomical Pathology, National Health Laboratory Service (NHLS), University of Pretoria, Hatfield 0028, South Africa; meshack.bida@nhls.ac.za; 4Institute for Cellular and Molecular Medicine, Department of Immunology, SAMRC Extramural Unit for Stem Cell Research and Therapy, Faculty of Health Sciences, University of Pretoria, Pretoria 0001, South Africa; chrisna.durandt@up.ac.za; 5Center of Systems Biology, Biomedical Research Foundation Academy of Athens, 4 Soranou Ephessiou Str., 115 27 Athens, Greece; achatzi@bioacademy.gr

**Keywords:** immunosuppression, immune evasion, PI3K pathway inhibitors, precision medicine, cancer cells

## Abstract

Immune response has been shown to play an important role in defining patient prognosis and response to cancer treatment. Tumor-induced immunosuppression encouraged the recent development of new chemotherapeutic agents that assists in the augmentation of immune responses. Molecular mechanisms that tumors use to evade immunosurveillance are attributed to their ability to alter antigen processing/presentation pathways and the tumor microenvironment. Cancer cells take advantage of normal molecular and immunoregulatory machinery to survive and thrive. Cancer cells constantly adjust their genetic makeup using several mechanisms such as nucleotide excision repair as well as microsatellite and chromosomal instability, thus giving rise to new variants with reduced immunogenicity and the ability to continue to grow without restrictions. This review will focus on the central molecular signaling pathways involved in immunosuppressive cells and briefly discuss how cancer cells evade immunosurveillance by manipulating antigen processing cells and related proteins. Secondly, the review will discuss how these pathways can be utilized for the implementation of precision medicine and deciphering drug resistance.

## 1. Introduction

Studies pertaining to the role of genomic instability in immuno and chemotherapeutic response are still a topic of interest, particularly in colorectal cancers [1,2,3]. This is due to the heterogenicity of these mutations within the different tumor microenvironments. Molecularly targeted therapies have been developed to target or block signaling pathways specific to a certain cancer type. This allows cancers to be sensitized to chemotherapy [4] or immunotherapy [5]. Despite the efforts to halt cancer progression at the DNA level, cancer can still persist and develop the ability to manipulate and evade the immune system. Cancer cells utilize various mechanisms to proliferate and survive. These cells take advantage of the normal functioning immunoregulatory processes and their related biochemical pathways to create a suitable environment for them to survive and thrive. These include the activities of immunosuppressive cells and the action of cytokines such as interleukin (IL)-10 and transforming growth factor-β (TGF-β) [6]. Immune checkpoints (ICs) are surface proteins that are crucial elements of immune regulation. They are characterized into stimulatory signaling pathways, which include glucocorticoid-induced tumor necrosis factor family-related protein (GITR) and T cell receptors (TCRs), and inhibitory signaling pathways, which involve cytotoxic T-lymphocyte-associated protein 4 (CTLA-4) and programmed cell death protein 1 (PD-1), amongst others [7]. Antibodies against these ICs have and are still being developed as cancer immunotherapies, but the efficacy of these treatments is hindered by immunosuppressive cells known as myeloid-derived suppressor cells (MDSCs), which stimulate another set of immunosuppressive cells known as regulatory T cells (Tregs) [8]. Similar to other immune cells, immunosuppressive cells are regulated by key signaling pathways. Such pathways are targeted for therapeutic purposes leading to possible cancer regression. However, in some instances, cancer might return even more aggressive due to the development of new mutations within the biochemical pathways resulting in possible drug resistance. It is worth noting that drug resistance through signaling pathways might be conferred by pathway reactivation which can take place via pathway rerouting or cross talk between interrelated signaling pathways [9]. The aim of this review is to highlight immunosuppressive pathways that could be targeted for therapeutic purposes and possibly find ways to decipher drug resistance through combinatorial targeted therapies.

## 2. Immune Evasion

Cancer cells are able to hijack the immune system by secreting cytokines and molecules familiar to effector T cells enabling them to evade immunosurveillance. Immunosurveillance is a process whereby the immune system guards against and averts cancer progression. Immunosurveillance is a concept that was first hypothesized by Paul Ehrlich [10] in 1909 when he proposed that the immune system restricted the growth of carcinomas. Five decades later, Burnet FA [11] and Thomas L [12] presented theories that supported Ehrlich’s theory. Burnet suggested that there might be a tumor-specific immune response that attempts to destroy developing cancer, whilst Thomas thought that there must be a mechanism similar to that of the host immune system versus foreign tissue, commonly seen in graft rejection, in which cancer can be fought off by the immune response. Even though this theory was proven correct, cancer cells still had a way of progressing, and that is when the concept of tumor immunoediting was hypothesized. This concept is divided into three different phases: elimination, equilibrium, and escape [13,14].

### 2.1. Elimination

The antitumor immune response is initiated by the activation of the innate immune system in the presence of cancer cells. Cells of the innate immune system are altered to favor proangiogenic activities and an immunosuppressive microenvironment. One of the major angiogenic factors, vascular endothelial growth factor (VEGF), limits tumor-infiltrating T cells and antigen-presenting cells (APCs) activity to foster immunosuppressive microenvironment through upregulation of Tregs and immune checkpoint inhibitors [15]. Exposure to carcinogens such as tobacco smoke or asbestos has been correlated with tissue disruption/inflammation through the activation of IL-1β, which enhances their tumorigenic ability [16]. Cancer cells stress promotes the production of other proinflammatory cytokines and proteins such as heat-shock proteins (HSPs) [17] and Natural killer group 2, member D (NKG2D), which serve as danger signals. Natural killer (NK) cells, macrophages, γδ T cells, and NK T cells are released to the tumor site resulting in cytotoxic effector mechanisms to eliminate cancer cells [18]. The release of interferon (IFN)-γ controls tumor growth and promotes the release of local chemokines allowing more innate immune cells to be recruited to the tumor site. If the tumor manages to grow beyond control, IFN-γ, seen as the “major effector of immunity” [19], is produced by the cells of the adaptive immune response poststimulation by tumor-specific antigens. In return, IFN-γ contributes to the stimulation of antitumor immune responses via a positive feedback mechanism [20]. The NK cells release reactive oxygen and nitrogen species which kill cancer cells via tumor necrosis factor (TNF)-related apoptosis-inducing ligand (TRAIL)-dependent or perforin dependent mechanisms, respectively, in an attempt to further eliminate cancer cells. The NK cells also promote differentiation and maturation of dendritic cells (DCs) via the production of cytokines, IL1, TNF-α, type I IFN, granulocyte-macrophage colony-stimulating factor (GM-CSF), and IL-15 [21,22,23].

### 2.2. Equilibrium

Equilibrium refers to a period between the failure of the immune system to completely eliminate/eradicate cancer cells and the beginning of the escape phase. This is the period where the malignant disease is clinically detectable. Cancer cells constantly adjust their genetic makeup either via nucleotide excision repair, microsatellite instability, or chromosomal instability, giving rise to new phenotypes that display reduced immunogenicity [24,25,26]. These cancer cells evolve with the generation of more advanced mutations that provide increased resistance to immunological attack until both immune and cancer cells are at an equilibrium state. These new variants of cancer cells have the ability to progress to the escape phase of the immunoediting process [22]. Furthermore, cancer cells induce alterations in the genome-processing mechanisms such as deoxyribonucleic acid (DNA) damage–repair machinery, telomere damage, centrosome amplification, and epigenetic modifications to develop new variants [27,28]. This is accomplished by hijacking processes that are mainly involved in cell division and tumor suppression [28]. Defective DNA damage repair results in the accumulation of immunosuppressive cells and decreased T cell responses [29]. These alterations have also been shown to contribute immensely to immunotherapeutic responses in cancer [30] and molecularly targeted cancer therapies. Of note, there are several key molecular signaling pathways identified that are associated with cancer progression and drug resistance. The most common of these pathways is the phosphatidylinositol 3-kinase (PI3K)/AKT/mTOR (tumor survival pathway) and its interrelated pathways, which will be discussed in more detail later in this review. The mechanism of action by inhibitors of this pathway includes the induction of DNA damage, particularly in cancers that take advantage of the DNA damage–repair system. Hence, dysfunctional production of nucleotides necessary for DNA synthesis and repair are the main components that allow for treatment efficacy with PI3K inhibitors (Figure 1). Interestingly, the AKT was shown to be less effective in eliciting DNA damage in cancer cells compared with PI3K [31], emphasizing the importance of targeting multiple pathways. One of the attempts to amplify immunological response in cancers is by activation of the cytosolic DNA sensor, cyclic-GMP-AMP synthase (cGAS)/stimulator of interferon genes (STING) pathway, which acts to detect cytosolic DNA and ultimately trigger an innate immune response by producing type I interferons (IFNs). A recent study showed that STING can elicit antitumor immune responses independent from type IFNs by recruitment of tank-binding kinase 1 (TBK1), essential for interferon regulatory factor (IRF) 3 activation [32]. Wayne et al. also explored DNA damage response pathways with the intention of activating an immune response in human cancer and improving therapeutic response to immune checkpoint inhibitors. The authors highlighted that the cytoplasmic DNA, whether single or double-stranded, increased TBK1 independently of pIRF3/7 or type I IFN response. Combination therapy with checkpoint kinase 1 ((CHK1) involved in the regulation of DNA damage repair and replication) inhibitors and chemotherapeutic drugs amplified cytoplasmic dsDNA compared with chemotherapy alone. This, however, caused a reduction in chemotherapy-induced IRF1 and the inability to activate type I IFN responses. The authors indicated that these findings might influence therapeutic strategies and decision making in combining immune checkpoint therapy with small molecule inhibitors [33]. DNA damage and cancer immunotherapy are discussed in more detail elsewhere [34]. Yaghmour et al. assessed whether the number and type of these alterations could be utilized as biomarkers of the effectiveness of immune checkpoint inhibitors in cancer patients with advanced disease. The authors noted that patients who were on immune checkpoint inhibitors and had a higher mutational burden showed significantly improved overall survival compared with their counterparts [35]. Smyth et al. postulated that investigating and understanding events that take place within the equilibrium phase in a controlled laboratory setting, as in the above study, might give rise to improved immunotherapeutic agents [21].

### 2.3. Escape

Cancer cells with an altered genetic makeup have the ability to withstand the immunological stress throughout the equilibrium stage and proceed to the escape phase, where they continue to grow without restrictions. The mechanisms utilized by cancer cells to proceed to the escape phase are attributed to their ability to alter antigen processing and presentation pathways.

#### 2.3.1. Escaping the Antigen Presentation Pathway

In a normal setting, tumor-associated antigens will be presented to cytotoxic T cells through major histocompatibility complex (MHC) class I; however, cancer cells downregulate the expression of proteins involved in antigen presentation including MHC I proteins and inhibit the maturation of DCs. This dispossesses cytotoxic T cells’ ability to recognize tumor cells, thus allowing them to evade immunosurveillance [36]. Downregulation or loss of MHC class I molecule expression could result from the heterogeneous expression of multiple tumor antigens that develop due to mutations in the β2 macroglobulin subunit [37,38]. Activated APCs destroy cancer cells by either engulfing them or through interaction with tumor-infiltrating NK cells. The NK cells’ method of destroying cancer cells can still be utilized to destroy cells with downregulated expression of MHC class I molecules [36]. To escape this, cancer cells develop other mechanisms such as downregulation of low molecular mass polypeptide (LMP) 2 and 3, which results in modifications of various antigens presented by MHC class I molecules [39]. Transporter associated with antigen processing (TAP) and tapsin proteins is responsible for loading antigen peptides onto MHC molecules. These proteins are mutated and downregulated, resulting in loss of MHC class I expression within the tumor microenvironment [40]. Cancer cells can also alter T cells’ response against tumor antigens developed during the selection period. This then poses a challenge for the immune system to recognize cancer cells as defective. Thus, the immune system ignores the presence of cancer cells by the downregulation of T cells’ response to tumor-specific antigens resulting in anergy [41]. The presence of tumor antigens can also cause abnormal differentiation of myeloid cells and DCs within the bone marrow via Jak2/STAT3 activation [42]. These immature myeloid cells accumulate in circulation and migrate to the tumor microenvironment, where they specifically suppress antigen-specific T cell responses [43,44].

#### 2.3.2. The PD-1/PD-L1 Pathway as a Mechanism of Escape

Programmed cell death protein-1 (PD-1) and its ligand PD-L1 can both be expressed on the surfaces of cancer cells, whilst PD-1 is predominantly expressed on the surface of immune cells. The PD-L1/PD-1 pathway’s primary function is the maintenance of immune tolerance and protecting the body from self-harm through the immunological attack. The downside of PD-L1/PD-1 pathways is that hindering T cells’ immune responses also provides a way for cancer cells to evade the immune system and survive. The immunosuppressive ability of this pathway encouraged the development of inhibitors against PD-L1/PD-1 proteins as cancer immunotherapy [45,46]. Juneja et al. studied the molecular mechanisms involved in the efficacy of PD-L1/PD-1 inhibitors. The authors noticed that the expression of PD-L1 on MC38 colorectal adenocarcinoma cells (sensitive to PD-1 inhibition) is enhanced in the presence of IFN-γ. The BRAF.PTEN and B16.F10 melanoma cells are apparently less sensitive to PD-1 inhibitors but also gave similar results indicating similarities in response despite the difference in cancer types. The PD-L1 protein was also shown to sufficiently act on its own in elucidating immune suppression. Antitumor immune responses are suppressed by blockade of cytotoxic T cells and maintenance of Tregs. Thus, inhibition of PD-1 enhances cytotoxicity T cells activity and production of related cytokines [47]. As with other cancer treatments, some cancers develop resistance to treatment with PD-1 inhibitors. Assessment of whole exon sequencing in patients with metastatic melanoma showed mutations related to IFN-receptor-associated Janus kinase (JAK) 1/2. The resultant loss of IFN-γ function needed for antitumor immunity (vide supra) enables cancer cells proliferation. Mutations in β-2-microglobulin resulted in the loss of MHC I surface expression [48]. This mutation is a common mechanism that cancer cells use to evade the immune system and develop immunotherapeutic resistance in melanoma [49]. The use of anti-PD-1 has since been used in several cancer types with more tolerable side effects than traditional cancer therapies, with more studies looking at strategies to assess its efficacy in combinatorial therapies [50].

## 3. The Tumor Microenvironment

The tumor microenvironment consists of a diverse population of nonmalignant cells/components, including immune cells, fibroblasts, stem cells, endothelial cells, secreted proteins, extracellular matrix, and blood vessels that can be manipulated by cancer cells to promote its proliferation and survival. Tumor cells accomplish this by establishing harmonious cross talk and interaction with the components of the tumor microenvironment (Figure 2). The components of the tumor microenvironment and the structure can differ according to cancer types [51,52]. Cells of the tumor microenvironment, particularly T lymphocytes, have been used in adoptive cell therapies either autologously or from allogeneic donors. Their use as a clinical diagnostic tool is well established, with more studies venturing on finding related biomarkers that can be used as predictors of patient clinical outcomes. One of these studies was performed by Forget et al., who suggested that patient stratification prior to immunotherapeutic treatment with CTLA4 or PD1 inhibitors is needed. The decision was based on finding ways to avert immunotherapeutic resistance and toxicity whilst identifying patients that would benefit most from tumor-infiltrating lymphocytes (TIL) treatment prior to and post-treatment with immune checkpoint inhibitors. The authors observed an improved treatment response in anti-CTLA-4 naive patients given a higher count of TIL that consists mainly of CD8 T cells. The CTLA-4 naive patients showed 24.6 months improved treatment response compared with 8.6 months in patients treated with CTLA-4 inhibitors [53]. Although the idea of defeating cancer using lymphocytes that have already developed anticancer mechanisms from the tumor microenvironment is promising, this has not been the case with respect to solid cancers. The tumor microenvironment exercises several measures to ensure drug resistance and its ability to grow. Tumor microenvironment adaptive drug resistance alludes to mechanisms unrelated to genetic or epigenetic changes utilized by the tumor microenvironment to resist cancer treatment. Amongst this is the dense environment that prevents penetration of drugs into the core of solid cancers. Intracellular signaling response elicited by tumor microenvironment factors plays a major role in therapeutic response [54]. Most times, solid tumors have hypoxic parts that make it difficult not only for drug delivery but also for the transport of nutrients/minerals and oxygen needed for cell survival due to disrupted vascularization resulting in poor blood supply (Figure 2).

Activation of hypoxia-inducible factor (HIF) signaling promotes cellular adaptation to hypoxic conditions. This gives cancer cells a unique phenotype that allows them to survive and grow beyond control with the ability to resist cancer therapy [55]. As one of the key factors that promote cancer progression, HIF regulates several processes within the tumor microenvironment. The mTOR pathway significantly induces HIF-1α. However, reduction in nutrition supply in cancer cells inhibits HIF-1α activity via a mTORC1 dependent mechanism [56]. The HIF-1/2α upregulates genes that assume control of immunomodulatory and metabolic processes within the tumor microenvironment. A process accomplished by induction of epithelial–mesenchymal transition (EMT)-related transcription factors such as SNAIL and ZEB family of transcription factors, and twist transcription factor (TWIST), to mention a few [57]. The EMT processes have also been implicated in cancer progression, metastasis, and drug resistance [58]. Of interest, the mTOR signaling pathway is associated with numerous immunosuppressive cells that contribute substantially to the development of a suitable tumor microenvironment for cancer progression and drug resistance. High mTORC1 activity and increased glycolytic metabolism were observed in effector Tregs similar to that seen in CD8+ T cells. Effector Tregs also showed higher activity of HIF-α and glycolysis enzymes, hexokinase 2 (Hk2), and phosphofructose kinase (Pfkp). Furthermore, effector Tregs mTORC1 activity was not comparable to their central Tregs counterparts during the analysis of glycolysis and TCA cycle metabolites. Overall, this data suggests that antigen exposed Tregs have high mTOR and glycolysis [59]. The mTOR and its regulation of immune response in the tumor microenvironment are discussed in more detail elsewhere. Here the authors also discuss the involvement of mTOR in the polarization of TAMs into the immunosuppressive M2 phenotype, which favors cancer progression and survival [60]. The mTOR pathway has also been shown to regulate cancer metabolism. Cancer cells metabolize glucose to produce adenosine triphosphate (ATP) as the source of energy without the need for oxygen. Because cancer cells grow at a rapid rate, they require more energy than the surrounding tissue. During this process, high levels of lactate are produced, an indication of anaerobic conditions. mTOR signaling pathway uses this metabolic activity to generate ATP and enhance cancer progression [61]. Cancer cells might also deprive antitumor cells within the tumor microenvironment of nutrients and energy. This could result in anergy of T cell responses, thus arming cancer cells with the ability to evade the immune system. Stable anaerobic conditions within the tumor microenvironment may also upregulate the mTOR signaling pathway, further ensuring cancer progression and survival [60]. This encouraged the development of mTOR inhibitors such as rapamycin analogs which function in association with FK binding protein (FKBP12). This interaction inhibits mTOR function and hinders cancer proliferation [62]. It is important to note that the mTOR signaling pathway is interrelated with the PI3K/Akt signaling pathways [60], which have been shown to play a key role in immunosuppressive signaling pathways.

## 4. Immunoregulatory Signaling Pathways

### 4.1. Myeloid-Derived Suppressor Cells

The MDSCs are a heterogeneous population of immature immune cells derived from a common myeloid progenitor within the bone marrow [63]. These cells will subsequently be differentiated into monocytes, macrophages, DCs, and granulocytes [64]. Their immunosuppressive effect is associated with the worst patient prognosis in cancer [65]. High levels of circulating MDSCs were also associated with the worst overall survival in patients with solid tumors suggesting the importance of these cells as potential therapeutic targets for the treatment of the disease [66]. They have also been implicated in reducing treatment response to immune checkpoint inhibitors [8]. Myeloid-related protein S100A9 has been shown to be one of the mechanisms that cancer cells use to block antitumor mechanisms. Overexpression of S100A9 increases the levels of MDSCs, which are associated with impaired maturation of APCs within the tumor microenvironment [67]. Multiple proteins are also involved in the regulation of MDSCs in cancer. Expressed mainly by hemopoietic cells and encoded by the *inpp5d* gene [68], SHIP-1 is a negative regulator of the PI3K/AKT downstream signaling pathway in a number of cellular activation processes, including myeloid survival. SHIP expression has been shown to be essential in the maintenance of myeloid cells. The downregulation of SHIP expression induced apoptosis in neutrophils and mast cells whilst downstream regulation of PI3K by AKT is associated with reduced apoptosis [69,70,71]. Neutrophils have been shown to promote cancer progress via degradation of insulin receptor substrate-1 (IRS-1). This result promotes PI3K interaction with mitogen platelet-derived growth factor receptor (PDGFR and resultant tumorigenesis [72]. In the absence of SHIP, AKT becomes phosphorylated, an effect abrogated by inhibition of PI3K, indicating its role as the key regulator of AKT [70] (Figure 3). The function of SHIP and its multiple immunomodulation of signaling pathways is discussed in more detail elsewhere [73]. The induction of PI3K activation is increased in myeloproliferative diseases such as acute myeloid leukemia [74]. Efforts to block PI3K along with mTOR signaling pathways for the implementation of molecularly targeted therapies in clinical trials are ongoing [75]. The parallel interaction between mTOR and PI3K signaling pathways implies their similarity in performing regulatory roles in cell proliferation and apoptosis needed for cancer survival and proliferation [76]. Thus, the use of SHIP alone or in combination with inhibitors of PI3K and parallel pathways in cancer has been postulated. One of the known signaling pathways that runs parallel with PI3K is mitogen-activated protein kinases (MAPK) signaling.

Chemotherapy targets rapidly dividing cells; hence it cannot differentiate between cancerous and noncancerous cells, leading to DNA damage in normal cells. Thus, more direct approach needs to be explored, and this includes finding ways to block cancer progression at a molecular level. Three of the main MAPK pathways were assessed for the role of MAPKs in MDSCs. Inhibitors of the ERK 1/2 and JNK pathways significantly increased the apoptosis of tumor and spleen-derived MDSCs [77]. Another study also showed that inhibition of PI3K/Akt along with the MAPK pathway, whilst ERK activation is retained, enhanced the doxorubicin-induced apoptosis of cancer cells [78]. The same effect was observed by Nair et al. when analyzing circulating MDSCs in colorectal cancer patients [79]. Genes expressed in the MAPK pathway were significantly increased in polymorphonuclear and monocytic MDSCs isolated from tumors than those isolated from spleens of the LL2 (xenograft lung cancer mice) tumor model. Inhibition of the MAPK pathway resulted in apoptosis and suppressed tumor growth [77]. Interaction between the MAPK pathway and GLI1 protein of the SHH pathway has been implicated in cancer progression in multiple cancers, and targeting a combination of these pathways could serve as another tool to decipher drug resistance to SHH inhibitors [80]. New therapeutic strategies for the inhibition of the PI3K pathway in a clinical setting are currently being investigated, and combinatorial therapeutic intervention with drugs such ibrutinib has been suggested [81]. Of note, SHIP has also been implicated in regulating the inhibitory cytokine, IL-10 signaling pathway [82] even in macrophages [83]. This cytokine facilitates STAT3/SHIP1 complex formation, which will later translocate to the nucleus in macrophages and induce the anti-inflammatory function of these cells [84]. This suggests that this complex can promote the polarization of macrophages into the protumorous M2 phenotype known to promote cancer progression.

### 4.2. Tumor-Associated Macrophages

Macrophages that infiltrate solid tumor microenvironments are referred to as tumor-associated macrophages (TAMs) [85]. These macrophages have been found in abundance in a number of cancers, including breast [86], colorectal [87], pancreatic [88], and prostate [89] cancers. High infiltration of macrophages within the tumor microenvironment is associated with reduced overall survival and treatment response. TAMs were found to be mainly of the M2 macrophage lineage, and their increased production of anti-inflammatory factors contributes to tumor progression [90]. The balance between TAMs and M1 macrophages is determined by signaling pathways such as the STAT pathway. The M1 macrophage polarizing signals induced by IFN-γ and lipopolysaccharide (LPS) activate the STAT1 pathway whilst STAT3/6 pathways are activated by M2 macrophage polarizing cytokines such as IL-10, IL-4, and IL-13 [91]. To tilt the scale towards the M2 phenotype, which assists in cancer progression, the kruepper-like 2 (KFL2) transcription factor, along with STAT6, induces M2 genes *Arg-1, Mrc1, Fizz1*, and *PPARγ*. In the same manner, M1 genes *TNF-α, Cox-2, CCL5*, and *iNOS* are blocked via the NF-κB/Hypoxia-inducible factor 1-alpha (HIF-1α) pathway [92]. Contradictory to these findings, STAT6 driven inhibition of the M2 polarization is achieved by Trim24 CREB-binding protein (CBP)-associated E3 ligase acetylation [93]. The activated STAT6 pathway can also induce the M2 phenotype via the IL-4 pathway, which is associated with lung cancer progression and is inactive in the M1 phenotype [94]. The switch from M1 to M2 macrophages is mediated by IRF/STAT signaling [95], while the LPS stimulated TLR4 will switch polarization towards the M1 phenotype. Thus, both the NF-κB/HIF-1α and IRF/TLR/STAT signaling pathways could be targeted in cancer to prevent cancer cells from persistently shifting macrophages into TAMs, which favor cancer progression within the tumor microenvironment. The PI3K/AKT signaling pathway seems to play a crucial role in immunoregulatory cells, including TAMs. By coculturing TAMs with lung adenocarcinoma cells, higher expression levels of PI3K/AKT proteins were observed [96]. The PI3K pathway has also been shown to be highly involved in macrophage polarization. The absence of PI3Kγ is associated with the M1 polarization [97], and activation of PI3K leads to M2 polarization [98]. Furthermore, JAK2/STAT3/STAT6 signaling pathways, along with other factors, have also been shown to favor M2 polarization [99] (Figure 4).

The Hedgehog signaling pathway has been shown to play a role in cancer progression [100]. Cancer cells secrete sonic Hedgehog (SHH) to promote their proliferation and survival. To do so, Hedgehog facilitates macrophage polarization within the tumor microenvironment into the protumor M2 phenotype [101]. Aberrant signaling of the Hedgehog pathway is associated with dysregulated tissue patterning and development, leading to a number of pediatric cancers as reviewed by Raleigh et al. [102]. Amongst these cancers is medulloblastoma (MB), which is the most common cancer that is predominantly treated with inhibitors of the smoothened (SMO) protein such as vismodegib or sonidegib [103]. The canonical SHH signaling pathway is initiated by the binding of SHH to PTCH 1. This binding leads to the inactivation of PTCH 1, which consequently downregulates smoothened SMO. At this point, SMO is translocated into the cilia membrane, a process required for the activation of GLI transcription factors and association with the negative regulator suppressor of fused (SUFU). The SUFU will then facilitate the translocation of GLI 2 & 3 into the nucleus [104] (Figure 5). Aberrant SHH signaling pathway leads to downstream activation of SMO; hence therapeutic approaches are focused on developing SMO inhibitors. The PTCH 1 protein is necessary for the inhibition of SMO. A recent study investigated the efficacy of SMO inhibitors in pediatric patients with somatic PTCH1 mutations. Although the use of SMO inhibitors in children impairs their normal growth and development, the authors suggested that these drugs, in combination with traditional therapies, can be used to promote long-term survival in these patients [103].

### 4.3. Regulatory T Cells

Regulatory T cells (Tregs) are known as one of the “master” immunoregulatory cells designed to maintain immune homeostasis. However, cancer cells take advantage of their suppressive effect on T cells to evade the immune response and continue to grow without restriction. Tregs are a heterogeneous set of immune cells. This means that finding a specific marker, particularly in humans, for inhibition strategies remains a challenge. Efforts to boost anticancer immune responses by blocking suppressive Tregs mechanisms are still being explored. This includes inhibition of Tregs-related suppressive cytokines and surface markers using antibodies [105]. For instance, the expression of the inhibitory IL-35 cytokine and chemokine receptors such as CCR5 recruits Tregs and activates AKT/mTOR signaling pathway to promote Tregs function [106]. The suppressive function of Tregs is known to be a contributing factor to cancer progression. The AKT/mTOR signaling pathway, generally known to be activated in cancer, is also responsible for cancer progression. Therapeutic interventions aimed at targeting these pathways have been implemented, with some being in clinical trials with positive patient responses [107]. Upon further investigation, cancer-related signaling pathways, P53 hypoxia, TNF receptor-associated factor 6–mediated (TRAF6-mediated), IFN regulatory factor 7 (IRF-7) activation, NKT pathway and inhibitory immune checkpoint receptors, T cell immunoreceptor with immunoglobulin and ITIM domains (TIGIT), PD-1, and cytotoxic T-lymphocyte-associated protein 4 (CTLA-4) were found to be upregulated in mice splenic Tregs [106]. Tregs also require the TCR signaling pathway to maintain their suppressive function. Treatment with Tamoxifen degraded the functional ability of the SLP-76 protein needed for T cells development. The authors noted that Tregs lacking this protein lost their suppressive effect, indicating the importance of this protein in the Tregs/TCR signaling pathway. Mutations in SLP-76, particularly Y145F, also abolished the suppressive function of Tregs [108]. This study suggests the need to further explore the use of Tamoxifen to disrupt immunosuppressive molecular signaling pathways in cancers other than breast cancer. Inhibition of the PI3K-Akt pathway, which is useful in TCR engagement and costimulation, results in reduced Tregs function and consequently poor cancer proliferation [109]. 

On the other hand, the efficacy of Staphylococcal Enterotoxin C2 (SEC2) as cancer immunotherapy by its ability to induce Tregs immunosuppressive function is questionable [110]. The SEC2 is a superantigen that connects MHC class II to TCR, resulting in hyperactivation of resting T cells [111]. This activation is accompanied by stimulation of cytokines such as GITR and programmed cell death protein 1 (PD-1), known to promote cancer progression (Figure 6). Therapeutic strategies to block PD-1 as potential cancer immunotherapy are being investigated [112]. Studies conducted on PD-1 positive Tregs indicated that these cells are capable of predicting the clinical efficacy of PD-1 blockade therapies better than its ligand, PD-L1, nor the mutational burden observed within the tumor microenvironment [113]. Selective inhibition of PI3K and PLCγ signaling pathways, which has been shown to play an important role in Tregs induction, significantly reduced the levels of SEC2-induced Tregs. IL-2 along with STAT5 phosphorylation proved to be important in the induction of Tregs by SEC2 [110]. The same effect was observed in IL-2/STAT5 signaling, which is needed for FoxP3 expression and the ultimate induction of Tregs [114]. The other pathway implicated in SEC2-induced Tregs is the TCR/NFAT/AP-1 signaling pathway. The balance between NFAT/AP-1 is crucial for effective immune responses. It has been indicated that a tilt in this balance could result in the inability to express inhibitory surface receptors, T cells exhaustion, and dysfunction [115]. The use of SECs in clinical trials has long been initiated, with minor modifications implemented over the years. This has been tested in phase I clinical trials in advanced pancreatic and colorectal cancers [116]. Phase II clinical trials were carried out in hepatocellular [117] and advanced renal cell cancers [118]. To date, SECs are still being explored and proven to be effective in other cancers, including bladder cancer [119]. The role of TGF-β in cancer has been thoroughly studied and established and can be described as a double-edged sword [120]. The potential use of the TGF-β/SMAD signaling pathway in deciphering drug resistance is still under investigation [121,122]. The stimulation of TGF-β needed for the expression of Tregs and their regulatory roles in the immune system is well studied [123]. Hence blocking the TGF-β/SMAD signaling pathway downregulates the SEC2-induced Tregs differentiation [110]. 

## 5. Personalized Precision Medicine and Combinatorial Therapies

Molecular testing platforms are employed to detect abnormalities between normal/cancerous tissue and the blood. Alterations in DNA, RNA, splicing factors, and post-translational modifications are used for diagnostic purposes, prediction of possible future development of disease mainly due to inheritance, and the development of therapies [124]. Molecularly targeted therapies are aimed at targeting these aberrant signatures either by down or upregulation of the genes relevant to a particular disease. Personalized precision medicine is aimed at developing individualized therapeutic strategies that are more effective at treating a specific type of disease with fewer adverse events and reduced therapeutic resistance [125]. Even though there is less toxicity and reduced adverse events compared with chemotherapy, targeted therapies have presented their own drawbacks in this regard. A recent study by Du et al. picked up a high incidence of adverse events in targeted therapies. The top five on their list included skin damage, fatigue, mucosal damage, hypertension, and gastrointestinal discomfort. The authors suggest that there should be more efforts dedicated to developing effective management strategies [126], and this should not only be focused on managing these adverse events, but attention should also be paid to the specificity and combination of therapies. Not all patients diagnosed with cancer benefit from molecularly targeted therapies. Patients must harbor specific cancer-related aberrant genes to receive treatment, and even so, the lifespan of these genes is reduced by constant mutations exerted by cancer cells resulting in drug resistance [127]. As a result, patient treatment options become limited, highlighting the need to find ways to decipher drug resistance.

A combination of molecularly targeted therapies with existing or newly developed chemoimmunotherapies should therefore be considered. All three immunosuppressive cells (MDSCs, TAMs, and Tregs) discussed earlier have a dysregulated PI3K signaling pathway as a common factor. Therapeutic interventions targeting PI3K are available, with some still in clinical trials (Table 1). However, the efficacy of PI3K inhibitors is limited by therapeutic resistance. Some of the methods suggested to overcome drug resistance are a reactivation of the PI3K signaling pathway in combination with parallel pathways such as (but not limited to) the AKT/mTOR signaling network and manipulation of the tumor microenvironment [105]. However, it was shown that the response rate can still be very low in some cases, such as treatment of triple-negative breast cancer patients with alterations in PIK3CA/AKT1/PTEN using buparlisib. Only three out of six patients with targeted DNA sequencing (MSK-IMPACT) had stable disease indicating the ineffectiveness of buparlisib in treatment in this cohort of patients [128]. A similar trend was seen by Rodon et al., who observed poor treatment response in colorectal cancer patients treated with buparlisib [129]. No difference was observed between patients receiving pictilisib and placebo in the phase 2 clinical trial that was performed in breast cancer patients with advanced diseased [130], yet again indicating the need for the development of combinatorial targeted therapies. Some of the approved PI3K inhibitors are already used as combinatorial therapies (e.g., Fulvestrant) in diseases such as advanced breast cancer [131,132]. The potential use of pictilisib in combination with doxorubicin which induced apoptosis in osteosarcoma cells resulted in inhibiting cancer progression [133]. The concept of using precision medicine where patients are treated based on a single gene mutation is not always efficient. Personalized combination therapies can be developed by targeting multiple related molecular genes/transcription factors/pathways in an effort to improve treatment response and overall survival.

Patients with follicular lymphoma are treated with an array of traditional cancer treatments either alone or in combination with other therapies such as radioimmunotherapy or chemoimmunotherapy. Failure of PI3K inhibitors as cancer treatment is attributed to several factors [137,138,139], including (but not limited to) mutations in members of the PI3K/AKT/NF-κB pathway that is due to NF-κB-induced resistance to apoptosis. Hence, drug resistance was abrogated by inhibition of this pathway in nasopharyngeal carcinoma (NPC) [140]. However, to reduce adverse events and drug resistance observed when this pathway is targeted, authors used a combinatorial therapeutic approach with indole-3-carbinol and silibinin in mice with lung cancer, which resulted in cancer reduction [141]. The PI3K/AKT/mTOR pathway hyperactivation also leads to the loss of sensitivity to endocrine therapy in breast cancer [142]. The most common alterations are seen in the loss of phosphatase and tensin homolog (PTEN) [143], known to inhibit the PI3K pathway. Angiogenic factors such as VEGF-1 have been shown to activate the PI3K pathway preferably in collaboration with PLCγ to promote cancer progression and migration [144] (Figure 7). The other component of the PI3K signaling pathway is PI3KCA which is correlated with cancer progression and drug resistance in the tongue [145] and breast cancers [146]. In an effort to develop combinatorial therapies to decipher drug resistance, Gupta et al. investigated the use of KIT/PI3K/MAPK (KPM) pathways as potential targeted therapy for gastrointestinal stromal tumors (GIST). The study found that inhibition of these pathways resulted in a sensitivity of GIST cell lines to treatment with imatinib. Treatment of imatinib-resistant cell lines with KPM significantly decreased the proliferation of these cells [147]. Inhibition of the PI3K/MAPK pathway was also associated with improved therapeutic response in doxorubicin-treated NIH3T3 cells [78]. Contrary to these findings, activation of the PI3K/MAPK pathway enhanced treatment response to the same drug-treated lung cancer cells [148]. Current studies are continuously making efforts to improve the efficacy of PI3K inhibitors by investigating interrelated signaling pathways. This includes the AKT/FoxO3a/PUMA pathway, whereby the induction of PUMA expression enhances copanlisib apoptotic activity in colorectal cancer cells. The study found PUMA to be the key mediatory molecule in regulating copanlisip-induced apoptosis, which can be used to decipher drug resistance or utilized as a possible biomarker of treatment response [149]. Another study found that using PI3K/MAPK inhibitors, copanlisib/refametinib in human epidermal growth factor receptor 2 (HER2)-positive gastric cancer alone or in combination with anti-HER2 therapy can significantly improve patient response to these therapies [150]. As expected, treatment with Idelalisib showed a reduction in the Tregs quantity and functional activity in patients with lymphomas [136], highlighting the effectiveness of molecularly targeted therapies in regulating the immune response in cancer patients. This is important as it has been indicated throughout the literature that immunosuppressive cells are mainly protumorous and related to reduced relapse time and overall survival in cancer patients.

To reduce adverse events and decipher drug resistance, multiple pathways and molecules associated with PIK3K are targeted. This includes inhibition of activated Akt/NF-κB with Silibinin and PI3K inhibitors to halt cancer progression. Alterations in PTEN results in activation of the PI3K pathway and lead to cancer progression; hence mutations in this pathway can be targeted to prevent PI3K activation. Angiogenic factors along with the PI3K pathway have also been shown to induce cancer progression. 
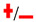
 positive/negative response.

## 6. Conclusions and Future Perspectives

Most cancer therapeutic endeavors are aimed at targeting a particular molecule at a time, but medicine is slowly revolutionizing, and multiple sets of proteins in combination with key molecular signaling pathways are being targeted for the development of more directed/personalized therapies. More and more studies are exploring the use of less invasive methods to diagnose or direct treatment decisions in cancer, and circulating immune cells have been at the forefront of these endeavors. Of note, SHIP, SHH, and SECs in MDSCs, TAMs, and Tregs, respectively, can be targeted in combination with the PI3K signaling pathway and related molecules in cancers where these cells serve as one of the prominent biomarkers of the disease or are associated with poor clinical outcome. In other instances where immunosuppressive cells serve as indicators of drug resistance, particularly in relation to immune checkpoint inhibitors, personalized combinatorial therapies can be explored to decipher drug resistance, including cases where treatment with PI3K inhibitors is ineffective. Realization of curative cancer strategies can be accomplished through tackling both immunological and molecular signaling pathways. To do so, researchers from different facets of cancer research can develop the ultimate cancer munition by combining specialties, as in the case of the use of immunotherapy or nanoparticle to enhance radiotherapeutic responses. Patient stratification can be performed through immunological and molecular biomarkers. This will assist in selecting patients who will better tolerate certain types of immunological treatment whilst exposing them to molecularly targeted therapies as well. An attempt to boost anticancer immunity while blocking the function of immunosuppressive cells can be achieved by blocking related molecular signaling pathways, which have also been shown to be activated in most cancers.

## Figures and Tables

**Figure 1 biomedicines-10-00682-f001:**
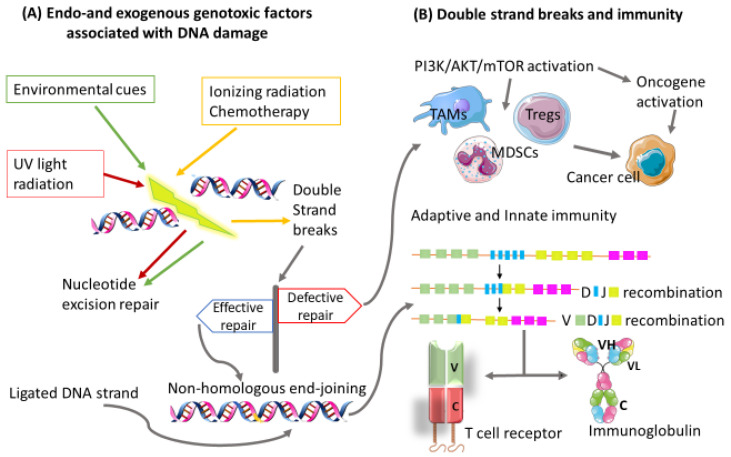
Impaired transcription and replication processes can occur, leading to DNA damage and genomic instability. (**A**) DNA damage can be a result of multiple factors, which could be of an endo or exogenous nature. (**B**) Repair of double-strand breaks (DSBs) is crucial to cell development and survival. DSBs are repaired without the need for a homologous template by the nonhomologous end-joining pathway. Consequently, an efficiently ligated DNA strand results in a heterogenous pool of antigen receptor genes needed for T and B cell development. Failure of the DSB repair mechanism activates the PI3K/AKT/mTOR signaling pathway, which induces immunosuppressive cells known to promote cancer progression. Variable region (v), Constant region (c), Heavy chain (H), Light chain (L).

**Figure 2 biomedicines-10-00682-f002:**
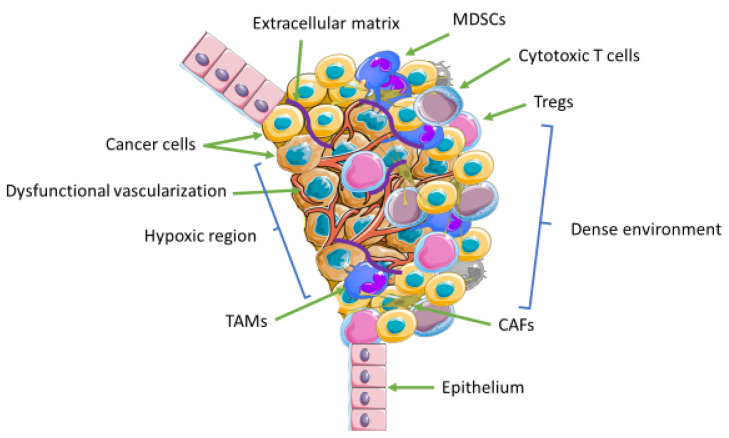
The tumor microenvironment consists of noncancerous cellular and noncellular components. There is a cross talk between cancer cells and the components of the tumor microenvironment, which cancer cells use to their advantage and create a suitable environment for cancer progression and survival. Amongst these is the recruitment of immunosuppressive cells, which dampens anticancer immune response and facilitates cancer growth and drug resistance. The dense regions of the tumor microenvironment prevent penetration of drugs into the core of the tumor. Blood vessels become cramped and dysfunctional resulting in a lack of blood supply and hypoxic conditions. Cancer cells use mechanisms that allow them to adapt and survive.

**Figure 3 biomedicines-10-00682-f003:**
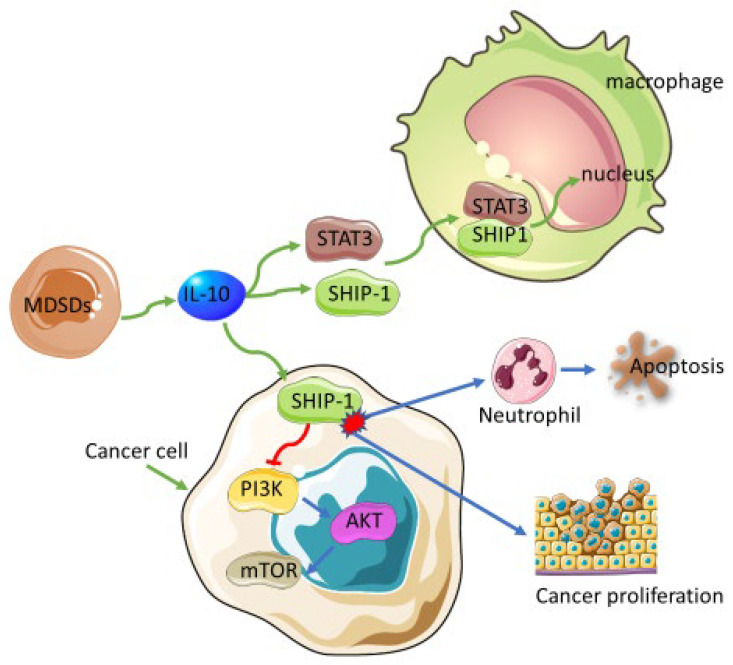
Tumor-associated MDSCs are the predominant suppliers of IL-10. The immunoregulatory effect of MDSCs and associated anti-inflammatory cytokines trigger the induction of the SHIP signaling pathway by cancers. This pathway is interlinked with the PI3K signaling pathway, which suppresses anticancer responses and promotes cancer progression via immune evasion. IL-10 also induces the anti-inflammatory effect of macrophages, thus potentially initiating their polarization into cancer-favoring M2 macrophages. Downregulated SHIP expression has also been shown to induce apoptosis in neutrophils needed for the destruction of cancer cells.

**Figure 4 biomedicines-10-00682-f004:**
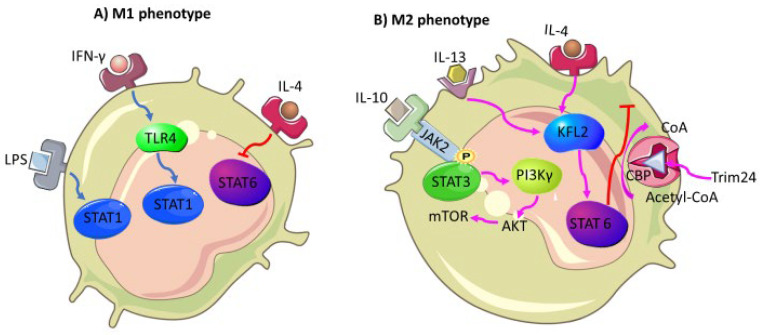
Molecular mechanisms favoring M2 polarization. STAT is one of the key signaling pathways with the prominent immunosuppressive capacity of tumor-associated macrophages. STAT6, in particular, serves as a double agent with its ability to favor cancer progression in collaboration with KFL2 but then again promote anticancer mechanisms via its gene machinery, which goes through acetylation by CBP to halt M2 polarization, a mechanism that can be targeted and manipulated so it can be activate long enough to allow the immune system to clear off cancer cells before they continue to the escape phase of immunoediting.

**Figure 5 biomedicines-10-00682-f005:**
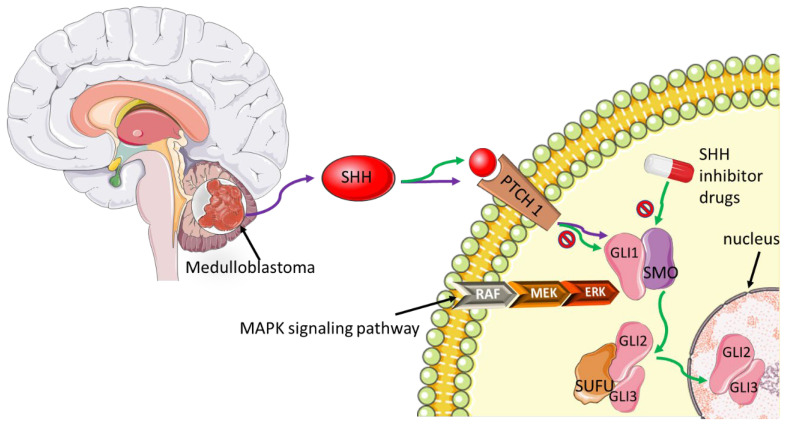
An overview of the SHH signaling pathway events in cancer. Efforts to block aberrant SHH signaling pathways induced by cancer cells which lead to downstream activation of SMO and cancer progression, have not been successful thus far. The mechanism of action by targeted inhibitory drugs such as vismodegib is through blocking the pathway at SMO. Multiple proteins associated with the SHH pathway and interrelated pathways (e.g., MAPK) could be targeted or used in a combinatorial therapeutic approach to decipher drug resistance to SMO inhibitors. 
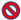
-Inhibition.

**Figure 6 biomedicines-10-00682-f006:**
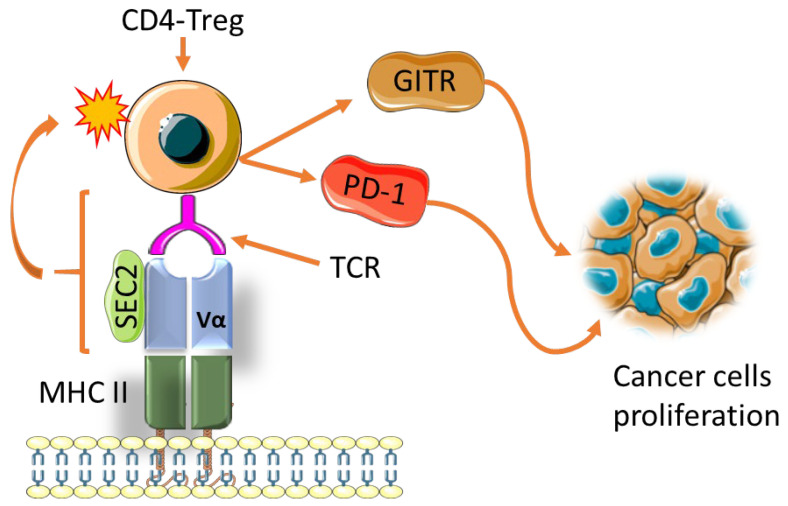
Enhanced immunosuppressive effect of SEC2-induced by activation of related cytokines serves as a barrier for effectively using these SEs as a treatment for cancer. Immunosuppressive Tregs are the major source of immune checkpoints such as GITR and also express PD-1. Tregs use these cytokines as one of the multiple ways to block anticancer mechanisms and aid in cancer cell proliferation and survival.

**Figure 7 biomedicines-10-00682-f007:**
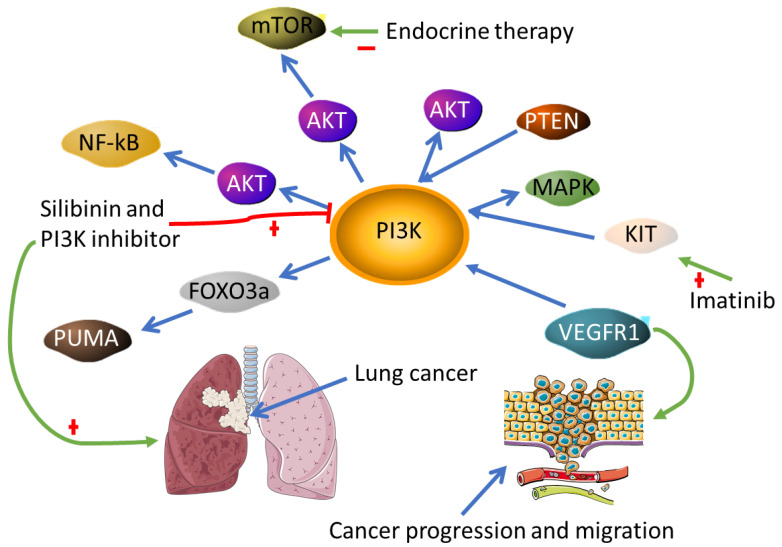
Factors involved in PI3K inhibitor therapeutic response in cancer.

**Table 1 biomedicines-10-00682-t001:** Examples of available PI3K inhibitors and their mode of action in different cancers.

PI3K Inhibitor	Mode of Action	Cancer Type	References
Alpelisib	PIK3CA/PI3K-δ isoform	Hormone receptor +/HER2-Breast Cancer	[131,132]
Copanlisib	PI3Kβ, PI3Kγ, PI3K-α & PI3K-δ isoforms	Follicular Lymphoma	[134]
Duvelisib	PI3K-δ and PI3K γ isoforms	Chronic Lymphocytic Leukemia	[135]
Idelalisib	PI3Kδ	Chronic Lymphocytic Leukemia	[136]
Buparlisib	PI3Kα, PI3Kβ, PI3Kδ and PI3Kγ	Metastatic triple-negative breast and colorectal cancers	[128,129]
Pictilisib	PI3Kα and PI3Kδ	Advanced breast cancer and cancer of the bone	[130,133]

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
