# Peer review of "Immunosuppressive Signaling Pathways as Targeted Cancer Therapies"

_biomedicines, 2022, doi:10.3390/biomedicines10030682_

Round 1

Reviewer 1 Report

In this review, the authors have provided a collection of information on the immunosuppressive signaling pathways and their therapeutic significance. This is a good review. However, the following points should be added to improve the manuscript,

  1. A section on the tumor microenvironment with a figure should be included.
  2. Information on the immunosuppressive metabolites in tumor immune evasion is needed.

Author Response

Dear Sir/Madam,

Thank you for taking time from your busy schedule to read the manuscript. It is highly appreciated.

See attached response.

Reviewer 2 Report

In this manuscript titled “Immunosuppressive Signaling Pathways as Targeted Cancer Therapies”, the authors first reviewed the basic concept of immune evasion and several immunoregulatory signaling pathways, then summarized the pros and cons of the current targeted cancer therapies featured by inhibition of PI3K signaling. Overall, this is an interesting topic and this review nicely summarized the current status of the development of targeted cancer therapies. However, this manuscript is poorly written, including extremely confusing figures and vague description of key information or concepts. Therefore, a major revision is needed.

Major issues:

  1. There is a large section describing how cancer cells “constantly adjust their genetic makeup using several mechanisms such as nucleotide excision repair as well as microsatellite and chromosomal instability” (Abstract, Line 106-114 and figure 1). How is this part related with targeted cancer therapies? Does any of the targeted therapies inhibit the genome alteration of cancer cells? This link is not explicitly described in this review. Actually, it has been shown that inhibiting PI3K can induce DNA damage (https://www.pnas.org/content/113/30/E4338) and inhibit DNA repair.

  1. Line 48, “Immune checkpoints 47 (ICs) are soluble surface proteins which are crucial elements of immune regulation”, what is the meaning of “soluble” here? Is it better to delete “soluble” here?

  1. The term GITR appeared at least twice in this manuscript, and its full name was written differently (line 50 and line 324). Please double check and ensure the accuracy of its full name.

  1. Line 81-82, “The antitumor immune response is initiated by the activation of the innate immune system in the presence of cancer cells and the resultant angiogenesis and tissue disruption”. This sentence is too vague and the following sentences didn’t explain it. For example, how does angiogenesis resulted from cancer initiate antitumor immune response?

  1. Line 89-90, “IFN-γ initiates the adaptive immune response via the activation of tumor specific antigens”. This sentence is very confusing, how does IFNg activate tumor specific antigens? By definition, tumor specific antigens are those antigens constantly expressed by tumor cells regardless of immune responses. Please elaborate.

  1. Line 148-149, “Activated antigen presenting cells (APCs) destroy cancer cells by either engulfing them or through interaction with tumor-infiltrating NK cells”, this sentence contains incomplete information which could mislead readers. The APCs also interact with T cells to destroy cancer cells, please elaborate this sentence by describing how APCs interact with other immune cells including T cells and NK cells.

  1. Line 158-160, “Autoimmunity is usually avoided during the development phase by the deletion of autoreactive antigens in the thymus, a process referred to as central tolerance”, this sentence here is very confusing. I don’t understand how this sentence connect with the sentences before and after it. This sentence looks like coming from nowhere.

  1. Line 176-179, the description of S100A9 is confusing. How does it relate to MDSCs? Such as do MDSCs produce S100A9, does S100A9 mainly work in MDSCs rather than other cell types?

  1. Line 198-200, “Inhibition of PI3K/Akt 198 along with the MAPK pathway whilst ERK activation is retained enhanced doxorubicin-induced apoptosis of cancer cells”, this sentence is very confusing. Is this describing the MAPK pathway inside MDSCs? How does MAPK pathway in MDSCs affect the apoptosis of cancer cells?

  1. There are two figure 2. Both are extremely confusing. Does the first figure 2 depict a cell? The title of this figure 2 is “Tumor associated MDSCs are the predominant suppliers of IL-10”, which is clearly different from the figure. The first figure 2 seems to suggest that IL-10 activate SHIP-1. The second figure 2 seems to show that M1 and M2 have different morphology, such as M2 has a smaller nuclear and bigger dendrite-like structure. Is this the case? Please re-draw these two figure 2 to clarify the above points.

  1. Line 225-226, “This increase in numbers has in most cases been associated with reduced relapse free, overall survival and treatment response”. This sentence is a little confusing, please rephrase to make it more concise.

  1. Line 297-302, this part is a little tedious and confusing. After reading it, I still don’t know the function of IL-35 and Treg in cardiovascular disease. Could this part be more concise? And how does this part relate to targeted cancer therapies?

  1. Line 319-332, please be more clear about whether SEC2 induces Treg or inhibits Treg. Details are needed to describe the current knowledge of how SEC2 is used as cancer immunotherapy.

  1. Figure 4 is extremely confusing too. In current figure 4, it seems that antigen presenting cells express GITR and CTLA4. And it is not clear whether the CD4 T cell is Treg or other CD4+ anti-tumor T cells.

  1. Line 351-352, “However, their success is limited by adverse events including nephrotoxicity that has been noted with these therapies”. This is not true. Adverse events exist in all kinds of cancer therapies, and the major issue specifically for immune checkpoint inhibitors is that they only work in a small proportion of patients in lots of cancer types.

  1. Generally speaking, I didn’t see a head-to-head or side-by-side comparison of targeting specific signaling pathways such as PI3K versus other cancer therapies in section “4. Personalized precision medicine and combinatorial therapies”.

Minor issue:

Line 100, “were” should be “where”?

Author Response

Dear Sir/Madam,

Thank you for taking time from your busy schedule to read and better the manuscript. This is highly appreciated.

See attached response.

Reviewer 3 Report

This review is a good attempt and covers the topic reasonably well. It is a thriving focus and covers all related issues In Immunosuppressive Signaling Pathways, which I enjoyed reading. Figures are well dispersive and very well fitted in the review topics. The authors have a good track record of having worked in this field. I recommended accepting this manuscript with minor comments.

  • It is better if authors include more information about tumors escape via PD-1-PDL1 interaction and pathways affected.
  • It will also be better to add some perspectives part in Concussion and retitle it Conclusions and perspectives.

Author Response

Dear Sir/Madam,

Thank you for taking time from your busy schedule to read the manuscript. Your contribution in this regard is greatly appreciated.

See attached response

Round 2

Reviewer 1 Report

The authors have addressed all the comments and the manuscript can be accepted for publication.

Author Response

Dear Reviewer,

Kindly see attached.

Thank you and wish you all the best in your career path.

Reviewer 2 Report

All major concerns are resolved. 

Only minor formatting or grammar issues need to be noted.

Minor issues:

In the pdf version of the manuscript, line 531, there is a star mark which may need to be deleted.

Line 566, "However, a showed that the response rate can still be very low...", it was showed?

Author Response

Dear Reviewer,

Kindly see the attached.

Thank you and wish you all the best in your career path.

RESPONSE TO REVIEWERS

Dear Editors,

Thank you once again for reading this manuscript titled; Immunosuppressive Signaling Pathways as Targeted Cancer Therapies.

Reviewer 2:

  • In the pdf version of the manuscript, line 531, there is a star mark which may need to be deleted.

Response

The star is intended to explain activation indicated in the diagram. It just needs to be aligned well during publication. It serves the same purpose as the – and + in figure 7.

  • Line 566, "However, a showed that the response rate can still be very low...", it was showed?

Response

Corrected.